# Effect of Liposomal Encapsulation and Ultrasonication on Debittering of Protein Hydrolysate and Plastein from Salmon Frame

**DOI:** 10.3390/foods12040761

**Published:** 2023-02-09

**Authors:** Kartik Sharma, Krisana Nilsuwan, Lukai Ma, Soottawat Benjakul

**Affiliations:** 1International Center of Excellence in Seafood Science and Innovation, Faculty of Agro-Industry, Prince of Songkla University, Hat Yai 90110, Thailand; 2Guangdong Provincial Key Laboratory of Lingnan Specialty Food Science and Technology, College of Light Industry and Food, Zhongkai University of Agriculture and Engineering, Guangzhou 510225, China

**Keywords:** protein hydrolysate, plastein, liposomes, encapsulation efficiency, bitterness, sonication

## Abstract

The impacts of liposomal encapsulation on the bitterness of salmon frame protein hydrolysate (SFPH) and salmon frame protein plastein (SFPP) with the aid of ultrasound (20% amplitude, 750 W) for different time intervals (30, 60 and 120 s) were investigated. Liposomes loaded with 1% protein hydrolysate (L-PH1) and 1% plastein (L-PT1) showed the highest encapsulation efficiency and the least bitterness (*p* < 0.05). Ultrasonication for longer times reduced encapsulation efficiency (EE) and increased bitterness of both L-PH1 and L-PT1 along with a reduction in particle size. When comparing between L-PH1 and L-PT1, the latter showed less bitterness due to the lower bitterness in nature and higher entrapment of plastein in the liposomes. In vitro release studies also showed the delayed release of peptides from L-PT1 in comparison to the control plastein hydrolysate. Therefore, encapsulation of liposomes with 1% plastein could be an efficient delivery system for improving the sensory characteristics by lowering the bitterness of protein hydrolysates.

## 1. Introduction

Peptides and food protein hydrolysates are regarded as a promising class of functional food ingredients. Due to their higher nutritional value, lower osmolarity and diverse bioactivities (e.g., antioxidant, anti-ageing, anti-hypertensive and immunomodulating activities), protein hydrolysates rich in bioactive peptides have recently gained popularity [1,2]. However, the development of bitterness occurs via enzymatic hydrolysis as a result of the release of hydrophobic domains from peptides. This directly restricts the use of protein hydrolysates for dietary purposes [3]. In addition, the limited bioavailability, hygroscopicity and propensity to interact with the food matrix of protein hydrolysates can prevent their commercial utilization [4]. Although numerous methods for reducing the bitterness of protein hydrolysates have been proposed, some drawbacks including the loss of hydrophobic amino acids, an excess production of free amino acids that result in a high osmolarity and low yields have been documented [5].

Plastein is a reaction, which has been used for lowering the bitterness of protein hydrolysate. Recently, Sharma et al. found that the use of papain at 0.1–1% for hydrolysis, followed by condensation and additional treatment with papain at 1% at 40 °C for 10 h was a potential plastein reaction, which could lower the bitterness of hydrolysates from salmon frame [3]. The encapsulation of protein hydrolysates in liposomes could be a potential method to form vesicles of active peptides [6]. Liposomes have been widely implemented in molecular biology, biochemistry, food science, medicine and pharmaceuticals [7]. Liposomes features depend on the material to be enclosed and stabilizing agents such as cholesterol, glycerol, etc. [8]. In general, cholesterol exhibits better properties when incorporated into liposomes, compared with glycerol [2,9]. However, considering the healthy dietary aspect, cholesterol is not desirable [10]. Liposomes upon formation can be single-layered (unilamellar) or multilamellar, having varying sizes [11]. Liposomes possess an amphiphilic structure with a wide range of characteristics, including biocompatibility, carrying capacity and protection of the loaded substances [12]. Moreover, liposomes are known for being biodegradable and non-toxic systems, and could mask the undesirable fish odor of hydrolyzed collagen from fish skin [2]. Vesicles with larger sizes possess lower absorption and have reduced stability upon storage. To overcome this problem, simultaneous use of other techniques such as use of ultrasound can be employed in size reduction of the vesicles [13]. Gulzar et al. also reported that the liposomes prepared using an ultrasound assisted process (UAP) had smaller sizes with higher encapsulation efficiency (EE) in comparison to those prepared using microfluidization [14]. Thus, ultrasonication can be used to improve the EE and stability of a liposome loaded with protein hydrolysates or plastein.

Peptides derived from fish proteins loaded into liposomes have improved antioxidant and antibacterial properties, as well as skin permeability [15]. Additionally, liposomal encapsulation could be a successful technique for lowering the bitterness of fish-derived hydrolysates. Nevertheless, no information on the use of liposomes for loading fish protein hydrolysate to prevent bitterness has been documented. In addition, the use of the plastein reaction in combination with entrapment in liposomes could be a potential means to reduce the bitterness. Thus, the present study aimed to lower the bitterness of the salmon frame protein hydrolysates and the plastein with the aid of ultrasonication. The resulting liposomes loaded with plastein prepared from salmon protein hydrolysate were also characterized and the release efficacy was also examined.

## 2. Materials and Methods

### 2.1. Chemicals

Glycerol and soy lecithin (phosphatidylcholine or SPC) were procured from Sigma-Aldrich, Inc. (St. Louis, MO, USA). Ethanol was purchased from Merck (Darmstadt, Germany) and acetone was obtained from RCIlabscan (Bangkok, Thailand).

### 2.2. Preparation of Salmon Frame Protein Hydrolysates and Plastein

Salmon frame protein hydrolysate (SFPH) and plastein (SFPT) were prepared [3]. Hydrolysis was carried out at 40 °C using papain at a concentration of 1% for 5 min to prepare the SFPH. For the SFPT, SFPH at a concentration of 30% was subjected to condensation and rearrangement for 10 h using 1% papain. Both SFPH and SFPT were freeze-dried, placed in zip lock bag and stored at −20 °C until further use.

### 2.3. Preparation of Liposomes

SFPH- and SFPT-loaded liposomes were prepared by the thin film hydration method as outlined by Chotphruethipong et al. [2]. SPC was mixed with glycerol, a stabilizing agent, at a 4:1 molar ratio. The mixture was dissolved in 20 mL of absolute ethanol at 50 °C to ensure complete solubilization and to attain a final concentration of 50 µM. With the aid of a rotary evaporator, the ethanol was evaporated at 55 °C until a thin layer was formed in a round bottom flask. The flask was then kept in a desiccator overnight to ensure complete removal of the ethanol. The resulting lipid films were dispersed in 20 mL of SFPH solution at different concentrations (1, 2 and 3%). The resulting liposomes loaded with 1, 2 and 3% SFPH were named L-PH1, L-PH2 and L-PH3, respectively. SFPT at varying concentrations (1, 2 and 3%) was also loaded into liposomes. Liposomes loaded with plastein at 1, 2 and 3% were referred to as L-PT1, L-PT2 and L-PT3, respectively. The mixtures were continuously stirred for 10 min. Empty liposomes were also prepared by the addition of water and was named as ‘EL’. Sonication using an ultrasonication bath (Elmasonic S 10H, Elma, Singen, Germany) was performed for 30 min to obtain uniform liposomes. The temperature was controlled at 30 °C by the addition of flaked ice into the water bath.

### 2.4. Characterization of SFPH and SFPT Loaded Liposomes

The SFPH and SFPT liposomes prepared using SPC–glycerol loaded with SFPH and SFPT at all concentrations were studied for the characteristics as shown below.

#### 2.4.1. Encapsulation Efficiency (EE)

The EE of the liposomes was determined using the method from Chotphruethipong et al. [2] with modifications. To 0.5 mL of freshly prepared liposomes, 1 mL of acetone was added and the mixture was centrifuged for 30 min at 5000× *g*. The supernatant containing unencapsulated SFPH or SFPT was collected, and the acetone was evaporated using an oven at 60 °C (Memmert, Schwabach, Germany). The residue obtained was resuspended in 2 mL of distilled water and used for the determination of peptide content using the Biuret method [16]. After mixing liposomes and 5% Triton X-100 at a 1:1 ratio (*v/v*), the mixture was vortexed until complete solubilization of SPC was achieved. Total peptide content was determined. EE was calculated using the following equation:

EE (%) = (Total amount of initial peptide − amount of unencapsulated peptide)/(Total amount of initial peptide) × 100

#### 2.4.2. Particle Size, Polydispersity Index and Zeta Potential

Freshly prepared liposomes were subjected to determination of particle size, polydispersity index (PDI) and zeta potential following the procedure of Chotphruethipong et al. using the dynamic light scattering technique with the aid of a ZetaPlus zeta potential analyzer (Brookhaven Instruments Corporation, Holtsville, NY, USA) [6]. All measurements were made with a medium refractive index of 1.333 at a temperature of 25 °C. After 120 s of autocorrelation, the samples (5 mL) were measured at a 90° angle to determine the particle size and PDI.

#### 2.4.3. Bitterness

Bitterness was determined as outlined by Sharma et al. [3]. The panelists participated in the training using caffeine as a reference twice a week for a period of one month. Caffeine standard solutions at various concentrations (0, 100, 200, 300, 400 and 500 ppm) were used, where distilled water represented a score of 0, while caffeine solution with 500 ppm concentration had a score of 15. For evaluation, a 15 cm line scale with the terms starting from “none” to “intense” was used. A random number was applied to code each sample before it was served to panelists. During the analysis, the panelists were advised to take the cracker and properly rinse their mouths using distilled water between each sample. The samples with the least bitterness were subjected to further analysis.

### 2.5. Ultrasonication and Characterization

#### 2.5.1. Ultrasonication of Liposomes Loaded with Selected SFPH and SFPT Concentrations

The liposomes loaded with SFPH or SFPT at the concentration yielding the least bitterness were selected for further ultrasound treatment using an ultrasonic probe (Sonics, Model VC750, Sonica% Materials, Inc. Newtown, CI, USA). The frequency and power output of the ultrasonic transducer were 20 KHz ± 50 Hz and 750 W, respectively. An iced water bath was used to maintain the sample temperature at 25 ± 5 °C. The samples were ultrasonicated for 30, 60 and 120 s at 20% amplitude with the pulse mode: 2 s pulse on and 8 s pulse off. The resulting liposomes were further characterized.

#### 2.5.2. Characterization of Liposomes

##### Particle Size, EE, PDI, Zeta Potential and Bitterness

All analyses were performed as detailed previously.

##### FTIR Spectra

Attenuated Total Reflectance or ATR-FTIR spectra of the freeze-dried liposome samples loaded with plastein showing the least bitterness and plastein were obtained using an FTIR spectrophotometer (Spectrum One, Perkin Ekmer, Norwalk, CT, USA) in the wavenumber range of 4000–600 cm^−1^. The spectrum of empty liposomes was recorded for comparison.

##### In Vitro Release Efficiency

The in vitro release efficiency of the liposomes loaded with plastein showing the least bitterness in comparison to plastein was measured using the method outlined by Hosseini et al. [17] with modifications. Briefly, a dialysis bag containing 2 mL of sample was placed in a beaker containing 50 mL of distilled water at room temperature while being continuously stirred (100 rpm). A UV-Vis spectrophotometer (Shimadzu, Tokyo, Japan) was used to measure the absorbance at 220 nm at designated time intervals (0, 2, 4, 8, 12, 24, 36 and 48). At each interval, 2 mL of the release medium was removed and replaced with an equal volume of fresh medium to maintain a constant volume.

##### Transmission Electron Microscopy (TEM)

Liposomes with the highest EE and the least bitterness were subjected to visualization of morphology with the aid of TEM as per the protocol of Tagrida et al. [18]. An aliquot of hydrolysate-loaded liposomes was placed on a complete membrane grid. The sample-loaded grids were stained with 2% uranyl acetate and were left to stand for 20 min. The grid was washed with sterile water and allowed to dry at room temperature (25 °C). The sample was visualized using a JEOL JEM-2010 transmission electron microscope (JEOL Ltd., Tokyo, Japan).

### 2.6. Statistical Analysis

Completely randomized design (CRD) was used for the whole studies. Experiments and analyses were performed in triplicate. Analysis of variance (ANOVA) and mean comparison were carried out using Tukey’s test with a 5% threshold for significance.

## 3. Results and Discussions

### 3.1. Characteristics of Liposomes Loaded with SFPH and SFPT at Varying Concentrations

#### 3.1.1. Encapsulation Efficiency (EE)

The EEs of liposomes loaded with SFPH and SFPT at various concentrations are listed in Table 1. The highest EE (*p* < 0.05) was observed when liposomes were loaded with 1% SFPH or 1% SFPT. However, the lowest EE was observed when SFPH or SFPT at 3% was loaded into liposomes. Generally, EE indicates the entrapment of the target substance inside the liposomes [19]. It is one of the most important parameters determining the capacity of loading the protein hydrolysate into liposomes. L-PH1, L-PH2 and L-PH3 showed EEs of 89.63, 69.49 and 60.49%, respectively. L-PT1, L-PT2 and L-PT3 possessed EEs of 99.98, 94.82 and 91.70%, respectively. The EE of liposomes decreased (*p* < 0.05) with increasing concentrations of SFPH or SFPT. This might be because the liposomes had restricted interior spaces. With excessive amounts of peptides, the vesicles could not uptake and keep all the peptides in their cores. A similar trend was observed by Chotphruethipong et al. [6], in which a lower EE was obtained as the concentration of the conjugates increased from 0.25% to 1% (*w/v*). These findings also correlated well with Mosquera et al. [20] who reported that liposomes likely became saturated at a specific peptide concentration and could not support an increased load of sea bream scale collagen. This phenomenon resulted in a decreased EE in the presence of excessive amounts of peptides. Zavaeze et al. [21] reported an EE of 80% upon loading of croaker proteins into soy lecithin liposomes, whereas Sarabandi et al. [22] documented an EE up to 90% when a flaxseed protein hydrolysate was loaded into lecithin liposomes. When comparing between liposomes loaded with SFPH and SFPT at the same concentration, the latter showed higher EEs (*p* < 0.05). Plastein products generally had lower hydrophobicity [3], thus allowing themselves to be trapped and able to interact with phosphate inside the core. The higher entrapment efficiency could be due to better placement of smaller peptides inside the core of liposomes [23]. Nevertheless, during thin film hydration, there are several factors such as raw material, chemical characteristics and ratios between the encapsulating materials and loaded compounds, which can determine EE [24]. Thus, EE was governed by the concentration of peptides in the protein hydrolysates, where the EE tended to decrease with increasing concentration of the peptides.

#### 3.1.2. Particle Size, PDI and Zeta Potential (ZP)

The particle size, PDI and ZP of liposomes loaded with SFPH and SFPT are given in Table 1. No difference (*p* > 0.05) in particle size between empty liposomes and liposomes loaded with 1% SFPH (L-PH1) was observed. In addition, when liposomes were encapsulated with 2% SFPH (L-PH2), a significant decrease (*p* < 0.05) in the size of the liposomes was observed. The decreased size could be attributed to the ability of protein hydrolysates, especially the lipophilic peptides or hydrophobic peptides, to increase cohesion packing between the apolar domains in membrane vesicles. The highest particle size was observed upon the entrapment of 3% SFPH (L-PH3). The formation of several layers of phospholipids with the insertion of free or unencapsulated hydrolysates into the layers might led to an increase in the size of the liposomes. The results were in line with those obtained by Varona et al., Moghimipour et al. and Detoni et al., where they observed an increased liposome size due to multi-lamellarity or flocculation [25,26,27,28]. Upon encapsulation of 3% SFPH, numerous free unencapsulated charged peptides would neutralize the charge of the liposomes, thereby resulting in the association of liposomes and higher agglomeration. Generally, the particle size is dependent on various factors such as the content of the phospholipids used during preparation, the method of preparation and the number of lamellae formed [29]. For liposomes encapsulated with SFPT, variations in particle size were observed when incorporated with SFPT at different concentrations. The lowest particle size was observed for L-PT1 (*p* < 0.05), whereas the highest particle size was observed for L-PT2 (*p* < 0.05). The particle sizes of SFPT-loaded liposomes with the optimal SFPT concentration (L-PT1) were smaller in comparison to those of SFPH-loaded liposomes having the optimized SFPH concentration (L-PH2). Higher hydrophobic interactions between hydrophobic domains took place for plasteins, leading to increased hydrophilicity [3]. The electrostatic and hydrophobic interactions between phospholipids and peptides are likely to be responsible for the amount of the substances that can be trapped inside the core [17]. In addition, multi-lamellarity may account for the higher particle size for L-PT2, while the EE was decreased. Therefore, the core substances played a role in the EE and size of the liposomes.

PDI is a measure of uniformity of the particles in any suspension. It is also an important indicator for the homogenous distribution of vesicles. The lower the value of PDI, the higher the homogeneity of particles in the suspension is. Generally, PDI values lower or higher than 0.3 indicate the homogeneity and heterogeneity in the sample, respectively [30]. Based on the highest EE for each liposome, those having the highest EE were L-PH1 and L-PT1. Nonetheless, the latter had a lower PDI (*p* < 0.05) than the former. The latter was distributed more uniformly than the former. Overall, the PDI values varied from 0.005 to 0.303. This indicated the presence of a narrow range of particle size distribution. A PDI value close to 0.4 was observed for chitosan-coated liposomes loaded with melatonin [31].

ZP has been used to determine the electrical charge present on the surface of liposomes [17]. All samples including blank liposomes showed negative ZP values, demonstrating that the surfaces of the liposomes were negatively charged. The polar heads or phosphate groups on phosphatidylcholine are responsible for the negative charge of the liposome surface. Lu et al. also documented the overall negative charge for liposomes [32]. Among liposomes, varying changes were noticeable. L-PT1 showed the highest negative charge which might be associated with the small size of this sample. Generally, the smallest size is correlated with a larger surface area, thus allowing the phosphate groups to be exposed to the aqueous phase. This resulted in an increased negative charge. However, the ZP of blank liposomes (−63.89 mV) (*p* < 0.05) was the second highest negative charge among all the encapsulated liposomes. The decreased negative ZP of the encapsulated liposomes with increased protein hydrolysate concentration might be due to the positive charge of the hydrolysates, especially the free hydrolysates that remained unencapsulated, which tended to partially neutralize the negative charge of the liposomes, especially on the outer surface of the liposomes. In addition, the positively charged unencapsulated peptides surrounding the liposomes might decrease electrostatic repulsions, leading to lower exposure of the liposome surface. Chotphruethipong et al. also reported the overall charge alterations of liposomes caused by surrounding unencapsulated peptides in liposomes loaded with hydrolyzed collagen from defatted Asian sea bass skin [2]. On the other hand, negatively charged unencapsulated peptides might be responsible for increased electrostatic repulsions between liposomes, resulting in an increased net negative charge of the liposomes. Due to much higher negative ZP of liposomes mediated by phosphatidylcholine, the overall charge of the encapsulated liposomes remained negative. The liposomes with higher negative ZP values tended to have higher stability and therefore the collapse of the liposomes was prevented [21]. Moreover, it can also be concluded that plastein products were found to impact the arrangement of the liposome bilayer as well as the particle size in this study. As a consequence, the ZPs of liposomes loaded with plastein were different from those loaded with protein hydrolysate. Due to dominant negative charge of the liposomes loaded with both encapsulated hydrolysate as well as plastein, they possessed higher stability.

#### 3.1.3. Bitterness

The bitterness scores of SFPH, SFPT and their corresponding loaded liposomes along with empty liposome are given in Table 2. Entrapment of the peptides significantly reduced the bitterness of SFPH and SFPT, regardless of their concentration used for loading. For SFPH, the bitterness values for PH1, PH2 and PH3 were reduced by 42.10, 16.42 and 13.41%, respectively, after encapsulation. Encapsulation leads to entrapment of hydrophobic peptides from hydrolysates into the core of liposomes. Because of entrapment, these bitter peptides could not react with taste buds upon consumption, hence reducing perception of bitterness. Gong et al. and Rao et al. observed the reduced bitterness upon encapsulation of whey peptides and casein hydrolysates in liposomes [33,34]. Reduced bitterness was also observed for SFPT after loading in liposomes. The bitterness score was reduced by 57.14, 21.15 and 15.63% for PT1, PT2 and PT3, respectively. The presence of wall materials made up of lecithin and glycerol inhibited the exposure of bitter peptides and some peptides might undergo hydrophobic interactions with the wall [35]. Generally, higher EE brought about higher entrapment of the peptides and reduced the bitterness. The synergistic effect of surface hydrophobicity and bitterness has been documented [4]. Since phospholipids have astringency to some extent, the ability of the panelists to assess the bitterness of the samples could be hampered to some degree. This was witnessed by some bitterness of empty liposomes detected by panelists. Overall, the least bitterness was attained in plastein (1%) loaded in liposomes (L-PT1), in which a score of 3.70 was noted. This was governed by negligible free plastein in the aqueous phase (EE = 99.98%). 

### 3.2. Effect of UAP on Characteristics of the Selected Liposomes Loaded with Protein Hydrolysate or Plastein

#### 3.2.1. Encapsulation Efficiency

The liposome samples with the least bitterness including L-PH1 and L-PT1 were subjected to UAP for different time intervals. The EEs of liposomes were decreased when liposomes were ultrasonicated (Table 3). However, the degree of the decrease was much higher in liposomes loaded with protein hydrolysate, whereas only a slight decrease was noted for liposomes loaded with plastein. The EEs of L-PH1 (89.63%) were decreased by 9.95, 13.85 and 15.90% when a sonication time of 30 s, 60 s and 120 s was applied, respectively. The EEs of L-PT1 (99.98%) were reduced by 0.14, 1.38 and 2.29%, when a sonication time of 30 s, 60 s and 120 s was applied, respectively. The decreased EE might be because of the cavitation effect of ultrasound. Turbulence caused by cavitation could be responsible for the disruption of liposomes, thus leaching peptides out from the membrane, resulting in a reduced EE. Prolonged sonication caused higher cavitation effects and higher decrease in EE was observed. Silva et al. also reported that several factors such as variation in amplitude, voltage and depth of probe had significant effects on the EE of the liposomes [36]. It was obvious that the reduction of EE was much less in liposomes loaded with SFPT than those loaded with protein hydrolysates. The hydrophilic domains of plastein localized in the core might interact with phosphate polar head of lecithin. As a consequence, such a network could maintain the core in the bilayer more effectively than liposomes with protein hydrolysate in the core.

#### 3.2.2. Particle Size, PDI and ZP

The particle size of the liposomes generally decreased as sonication time increased as shown in Table 3. For L-PH1, the particle size decreased by 51.44, 53.14 and 62.85%, when an ultrasonication time of 30, 60 and 120 s was used, respectively. For L-PT1, the percentage decrease was much less than those observed for liposomes loaded with protein hydrolysate. The particle size decreased by 1.06, 4.97 and 18.43% for liposomes loaded with plastein and subjected to ultrasonication for 30, 60 and 120 s, respectively. The decreased particle size could be due to the efficient conversion of larger multilayered liposomes into unilamellar ones under the ultrasonic treatment. Usually, the particle size is dependent on the number of layers or lamellae of the liposomes. These lamellae may vary from 20 nm to several mm in thickness [37]. Similar observations for reduced particle sizes of liposomes were reported by Silva et al. [36]. A short sonication time was required to obtain nano-sized vesicles. Excessive treatment time might rupture the vesicles, resulting in reduced EE as previously discussed. In addition, the lower percentage decrease for the L-PT1 samples could be related to the resistance of the lamellae which were tightly interlinked between plastein and the wall, especially hydrophilic peptides adhering to the wall internally. This resulted in higher encapsulation efficiency.

The PDI values decreased significantly with increasing sonication time. Among the liposomes loaded with protein hydrolysate, the lowest PDI value was recorded for an ultrasonication time of 120 s (0.005) (*p* < 0.05). A similar result was observed for liposomes loaded with plastein. As stated earlier, a PDI less than 0.3 indicates the mono-dispersity or homogeneity of the suspension. Therefore, the results revealed that higher uniformity of liposome particles could be achieved by ultrasonication for a longer time. Lower PDI values also point towards the stability of the liposomes [29]. Overall, the values remained below 0.3 for all samples, thereby indicating the homogeneity and stability of the liposomes, irrespective of ultrasonication time used.

All the samples showed negative ZP values (*p* < 0.05). The negative ZP of L-PH1 increased by 19.57, 13.62 and 13.64% when ultrasonicated for 30, 60 and 120 s, respectively. The increased negative surface charge or zeta potential after sonication might be due to the formation of unilamellar liposomes from the multilamellar ones. This coincided with a reduction in particle size. Sonication may have also helped in the dispersion of liposomes adhered to one another. This phenomenon could result in exposure of the lipid bilayer to expose all its negative charge, which was earlier covered by other layers of liposomes. However, for SFPT-L-PT1, the negative ZP decreased by 2.8, 2.96 and 3% after being ultrasonicated for 30, 60 and 120 s, respectively. The changes were quite small and no differences were found among treated samples (*p* > 0.05). This might be related to the potential maintenance of the core (plastein), regardless of ultrasonication time, reducing the collisions among different liposomes. However, some of the hydrophobic core could be released to small extent due to the cavitation effect, thus covering the negatively charged surface. This was a result of increased repulsive interactions among the liposomes. Hosseini et al. and Bourab et al. reported the effect of the presence of hydrophobic molecules in the bilayer that tend to mask the negative charge of the liposomes, especially on the surface [17,38]. Since the negative charge was above −30 mV for all the samples, all samples still had high stability and flocculation could be prevented.

#### 3.2.3. Bitterness

The ultrasound assisted process applied to liposomes resulted in increased bitterness as shown in Table 4. The bitterness score was increased as the ultrasonication time increased (*p* < 0.05). The bitterness scores were increased by 18.30, 27.23 and 40.63% for SFPH samples (*p* < 0.05) in comparison to those of the control (L-PH1) when the sonication time was 30, 60 and 120 s, respectively. For SFPT, the increase was 19.32, 30.68, 56.25%, in comparison to those of the control (L-PT1) after 30, 60 and 120 s, respectively. The cavitation effect of ultrasound likely resulted in the disruption of liposomes, thus releasing the hydrophobic peptides from the cores of the liposomes. This resulted in increased bitterness perception. This could also be supported by the reduced EE upon increased sonication treatment time. Generally, the EE is inversely proportional to bitterness. The higher the EE, the less the bitterness was perceived [39]. When all the peptides could not be efficiently incorporated into the liposomes, parts of them might remain over the surrounding layers of liposomes. This contributes to the increased bitterness. Similar observations were documented by Rao et al. who showed the reduced bitterness upon encapsulation. Liposomal walls aid in hindering the exposure of casein hydrophobic peptides [34]. Between both controls, L-PT1 and L-PH1, the former showed a lower bitterness (*p* < 0.05). This could be due to the better entrapment of the hydrophobic peptides inside the liposome. In addition, plastein itself showed a lower bitterness score compared to protein hydrolysate [3]. Thus, ultrasound had a negative impact on sensory properties by enhancing the bitterness of liposomes loaded with salmon frame protein hydrolysates and plastein.

#### 3.2.4. FTIR Spectra

ATR-FTIR spectra of the freeze-dried samples involving empty liposome (EL), plastein (PT1) and liposomes loaded with plastein (L-PT1) are shown in Figure 1. Structural changes can be clearly seen in protein hydrolysate (PH1) at 1400–1600 cm^−1^, which represents C-N stretching vibrations. In addition, a slight change could be observed in the range of 1380–1459 cm^−1^, which is related to amide III stretching vibrations. The distinguishable peaks of PH1 from 1500 to 1650 cm^−1^ signifies C-O stretching vibrations of amide I and N-H stretching and deformation with C-H vibrations. EL and L-PT1 showed different peaks in the range of 2850–3000 cm^−1^, where L-PT1 showed a higher amplitude. These peaks are characteristic C-H2 asymmetric vibrations. These vibrations reveal a flexible acyl chain in the lipid membrane [40]. Moreover, the slight shift of a peak from 2924 cm^−1^ in the EL samples to 2921 cm^−1^ in L-PT1 could be due to ionic interactions among the peptides and phospholipids. After encapsulation, the peak at 1736 cm^−1^ of the empty liposome (EL), which results from the stretching C=O vibrations of the aliphatic stearic acid chain and the functional group, was shifted to 1742 cm^−1^, indicating the interaction between hydrogen bonds between carbonyl (C=O) groups of the phospholipid and the plastein [22]. The stretching PO_2_ vibrations are characterized by a band between 1040 and 1230 cm^−1^. The existence or absence of hydrogen bonds between phosphate groups and the hydrogen atom of bioactive compounds is indicated by higher and lower frequencies in this range, respectively [40]. The shift from 1058 cm^−1^ to 1050 cm^−1^ in the liposomes loaded with plastein indicated an interaction between the phosphate group of lecithin and the hydrophilic domain of plastein. Following the encapsulation of the casein hydrolysates, a similar behavior was seen [22]. The band at 970 cm^−1^ from the empty liposome can be attributed to the choline’s asymmetric stretching vibrations (N-(CH_3_)_3_) in the polar area of the phospholipid. This band was absent in PH1. However, in L-PT1 the band was shifted to 972 cm^−1^, suggesting that the peptides might localize to the polar area of lecithin (phosphatidylcholine). Similar bands were observed by Zavareze et al., where the bands with this wavenumber were formed in encapsulated fish protein hydrolysates [21]. Overall, the FTIR spectra suggested the interaction between the functional groups of the phospholipids and the side chains of the peptides in plastein.

#### 3.2.5. In Vitro Releasing Efficiency

The releasing efficiency (RE) of liposomes loaded with plastein (L-PT1) and free PT1 was measured as a function of time. Generally, in vitro release studies serve as a basis for in vivo research by allowing the screening and evaluation of various formulations according to the rates of compound release [17]. The oral delivery mechanism must release bioactive substances under controlled conditions for their potential advantages to be realized [17]. The release profile of encapsulated and free plastein is given in Figure 2. Rapid release of the unencapsulated peptides in the plastein was observed in comparison to those loaded in liposomes. Almost all peptides (approximately 95%) were released within 24 h, whereas the release was about 59% from the loaded liposomes. Similar observations were noted by Hosseini et al., who reported the slower release of peptides from fish gelatin hydrolysates upon loading in liposomes [17]. Peptides without encapsulation were completely released within 8 h; however, for encapsulated peptides, the release was only up to 40%. The delayed release could be attributed to the presence of wall like material or lamellae of the liposomes which slowed the escape of the peptidic fractions out from the liposomes. In addition, the negative charge on the walls of the liposomes due to lecithin tended to provide more stability to the liposomes, thereby decreasing the release of peptides. Usually, the type of liposome, composition of the lamellae and presence of a stabilizing agent had direct effects on the releasing rate [18]. Thus, encapsulation of plastein in liposomes significantly reduced the diffusion rate and controlled the release of peptides in plastein.

#### 3.2.6. Transmission Electron Microscopic images of Sample with Least Bitterness

TEM is extensively used to investigate the morphology of nano-sized particles along with their characteristics. When EL and L-PT1 were visualized under TEM, the liposomes showed a spherical morphology (Figure 3). It could be clearly observed that the size of liposomes loaded with plastein (L-PT1) (Figure 3B) was much smaller than that of EL (Figure 3A). This was in line with the particle size that was previously observed in this experiment. Furthermore, with increased magnification (245,000×), loaded liposomes with a double layer were found in L-PT1 (Figure 3C). Figure 3A,C depicts that EL had an empty core, whereas L-PT1 had the loaded plastein inside its core. For the empty liposomes (Figure 3A), some free lecithin could not form the bilayer structure and localized in the aqueous phase, appearing as the debris-like material surrounding the liposomes. For the liposomes loaded with plastein (Figure 3B), the liposomes were not contiguous. This is because most liposomes were negatively charged. Therefore, there was repulsion between different liposomes, thus keeping those liposomes separated from each other. It was noted that the liposome images in Figure 3C had a dark core. This was because the contrast was increased to see the outer layer clearly in combination with the higher magnification (245,000×) (Figure 3C). Overall, the image brightness generally decreased as magnification increased. The schematic representation of the localization of hydrophilic and hydrophobic peptides in the hydrolysate or plastein inside the core and their distribution in the liposome structure, both on the surface and between the bilayer, is shown in Figure 4. Thus, it can be concluded that the plastein was effectively loaded into the liposomes of phosphatidylcholine with glycerol as a stabilizing agent and possessed a spherical shape along with high stability.

## 4. Conclusions

Liposome encapsulation of SFPH and SFPT greatly influenced the particle size, encapsulation efficiency and overall bitterness of the hydrolysates. The concentration of SFPH and SFPT had a profound effect on EE and bitterness. Liposomes loaded with 1% plastein (L-PT1) showed the highest reduction in bitterness. However, the use of ultrasound resulted in lower EE and increased bitterness of liposomes loaded with either SFPH or SFPT. Therefore, encapsulation of plastein in liposomes could be employed to reduce the bitterness of fish protein hydrolysate, especially its plastein without the use of ultrasound. Since the plastein incorporated into liposomes still possessed some bitterness, further reduction of bitterness could be achieved by adding additional layers of liposomes in order to shield the surface of the liposomes. This might aid in using the hydrolysates in food at much higher concentrations to avail the benefits of hydrolysates in terms of nutrients and nutraceuticals.

## Figures and Tables

**Figure 1 foods-12-00761-f001:**
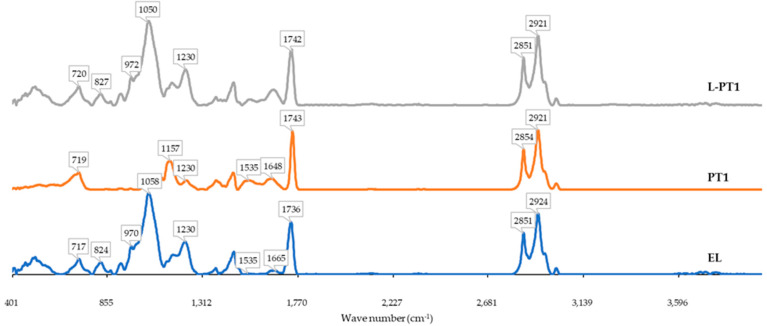
FTIR of empty liposome (EL), plastein hydrolysate (PH1) and plastein-loaded liposomes (Phe1).

**Figure 2 foods-12-00761-f002:**
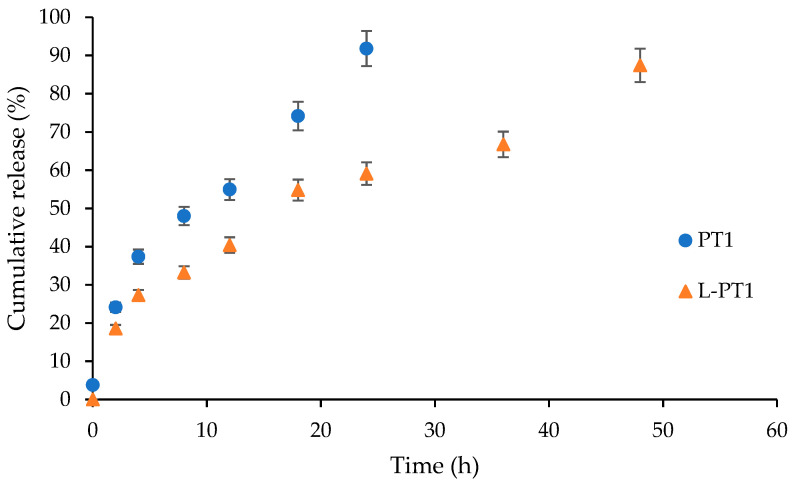
Release profile of plastein (PT1) and liposomes loaded with 1% plastein (L-PT1) prepared using thin film hydration during 48 h of dialysis at 28 ± 2 °C. Bars represent standard deviation (*n* = 3).

**Figure 3 foods-12-00761-f003:**
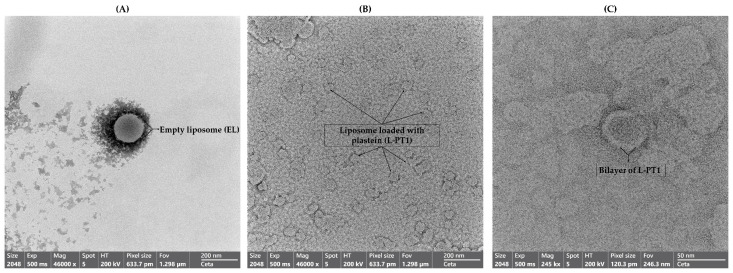
Transmission electron microscopic (TEM) image of (**A**) empty liposome (EL) at 46,000× magnification, (**B**) liposome loaded with 1% plastein (L-PT1) at 46,000× magnification and (**C**) L-PT1 at 245,000× magnification.

**Figure 4 foods-12-00761-f004:**
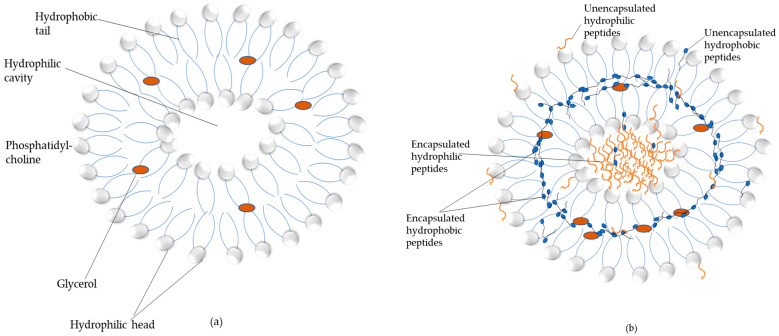
Schematic diagram representing the arrangement of lipids and proteins in an (**a**) empty liposome and (**b**) encapsulated liposome.

**Table 1 foods-12-00761-t001:** Encapsulation efficiency (EE), particle size (PS), polydispersity index (PDI) and zeta potential (ZP) of liposomes loaded with SFPH and SFPT.

Sample		Encapsulation Efficiency (EE) (%)	Particle Size (PS) (nm)	Poly Dispersity Index (PDI)	Zeta Potential (ZP) (mV)
	EL	-	546.0 ± 3.7 ^bB^	0.15 ± 0.04 ^bC^	−63.9 ± 1.2 ^aB^
SFPH	L-PH1	89.6 ± 0.3 ^a^	543.5 ± 1.0 ^b^	0.30 ± 0.02 ^a^	−59.5 ± 1.0 ^b^
	L-PH2	69.5 ± 1.2 ^b^	392.7 ± 3.0 ^c^	0.22 ± 0.01 ^b^	−51.7 ± 0.73 ^c^
	L-PH3	60.5 ± 1.4 ^c^	859.0 ± 7.2 ^a^	0.25 ± 0.00 ^c^	−42.0 ± 1.2 ^d^
SFPT	L-PT1	99.9 ± 0.0 ^A^	283.8 ± 3.3 ^C^	0.26 ± 0.04 ^B^	−68.2 ± 1.2 ^A^
	L-PT2	94.8 ± 0.0 ^B^	659.9 ± 14.5 ^A^	0.27 ± 0.00 ^A^	−48.8 ± 0.9 ^C^
	L-PT3	91.7 ± 0.6 ^C^	549.0 ± 8.6 ^B^	0.01 ± 0.00 ^D^	−39.1 ± 1.4 ^D^

Note: EL: empty liposome, SFPH: salmon frame protein hydrolysate, L-PH1: liposomes loaded with 1% protein hydrolysate, L-PH2: liposomes loaded with 2% protein hydrolysate, L-PH3: liposomes loaded with 3% protein hydrolysate, SFPT: salmon frame protein plastein, L-PT1: liposomes loaded with 1% plastein, L-PT2: liposomes loaded with 2% plastein, L-PT3: liposomes loaded with 3% plastein. For EE, the different lowercase or uppercase superscripts in the same column within the same sample (SFPH or SFPP) indicate significant differences (*p* < 0.05). For PS, PDI and ZP, different lowercase or uppercase superscripts in the same column within the same sample (SFPH or SFPP) including empty liposome indicate significant differences (*p* < 0.05).

**Table 2 foods-12-00761-t002:** Bitterness of liposomes loaded with SFPH and SFPT.

Sample		Bitterness
	EL	6.5 ± 0.7 ^deBC^
SFPH	PH1	5.8 ± 0.4 ^e^
	PH2	8.7 ± 1.2 ^bc^
	PH3	10.2 ± 0.9 ^a^
	L-PH1	4.2 ± 0.4 ^f^
	L-PH2	7.6 ± 1.4 ^cd^
	L-PH3	9.1 ± 1.2 ^ab^
SFPT	PT1	5.3 ± 0.4 ^C^
	PT2	7.7 ± 1.1 ^AB^
	PT3	8.9 ± 0.8 ^A^
	L-PT1	3.7 ± 0.4 ^D^
	L-PT2	6.6 ± 1.4 ^BC^
	L-PT3	7.9 ± 1.0 ^AB^

Note: Different lowercase or uppercase superscripts in the same column within the same sample (SFPH or SFPP) including empty liposome indicate significant differences (*p* < 0.05). For bitterness, scores are based on 15 cm line scales (0: none and 15: intense).

**Table 3 foods-12-00761-t003:** Encapsulation efficiency (EE), particle size (PS), polydispersity index (PDI) and zeta potential (ZP) of SFPH- and SFPT-loaded liposomes treated with ultrasound assisted process (UAP).

Loaded Sample	Ultrasonication Time (s)	Encapsulation Efficiency (EE) (%)	Particle Size (PS) (nm)	Poly Dispersity Index (PDI)	Zeta Potential (ZP) (mV)
L-PH1	0	89.6 ± 0.3 ^a^	543.5 ± 1.0 ^a^	0.30 ± 0.02 ^a^	−59.5 ± 1.0 ^c^
	30	80.7 ± 0.2 ^b^	263.9 ± 1.7 ^b^	0.17 ± 0.01 ^b^	−71.2 ± 0.9 ^a^
	60	77.2 ± 1.0 ^c^	254.7 ± 2.2 ^c^	0.11 ± 0.02 ^c^	−67.6 ± 0.4 ^b^
	120	75.4 ± 0.5 ^d^	201.9 ± 2.8 ^d^	0.01 ± 0.00 ^d^	−67.6 ± 0.3 ^b^
L-PT1	0	99.9 ± 0.0 ^A^	283.8 ± 3.3 ^A^	0.26 ± 0.04 ^A^	−68.2 ± 1.2 ^A^
	30	99.8 ± 0.4 ^A^	280.8 ± 2.8 ^A^	0.22 ± 0.01 ^A^	−66.3 ± 0.6 ^B^
	60	98.6 ± 0.4 ^B^	269.7 ± 0.6 ^B^	0.16 ± 0.11 ^A^	−66.2 ± 0.9 ^B^
	120	97.7 ± 0.4 ^C^	231.5 ± 1.0 ^C^	0.17 ± 0.02 ^A^	−66.2 ± 0.5 ^B^

Note: SFPH: salmon frame protein hydrolysate, L-PH1: liposomes loaded with 1% protein hydrolysate, SFPT: salmon frame protein plastein, L-PT1: liposomes loaded with 1% plastein. Different lowercase or uppercase superscripts in the same column within the same hydrolysate indicate *p* < 0.05.

**Table 4 foods-12-00761-t004:** Bitterness of SFPH (L-PH1)- and SFPT (L-PT1)-loaded liposomes treated with ultrasound assisted process (UAP).

Loaded Sample	Ultrasonication Time (s)	Bitterness
L-PH1	0	4.5 ± 0.0 ^cC^
	30	5.3 ± 0.7 ^bC^
	60	5.7 ± 0.8 ^abAB^
	120	6.3 ± 0.4 ^aA^
L-PT1	0	3.5 ± 0.04 ^cD^
	30	4.2 ± 0.6 ^bC^
	60	4.6 ± 0.2 ^bC^
	120	5.5 ± 0.3 ^aB^

Note: See Table 3 footnote. For bitterness, scores are based on 15 cm line scales (0: none and 15: intense). Different lowercase or uppercase superscripts in the same column within the same hydrolysate indicate *p* < 0.05.

## Data Availability

Not applicable.

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
