# Peer review of "Effect of Liposomal Encapsulation and Ultrasonication on Debittering of Protein Hydrolysate and Plastein from Salmon Frame"

_foods, 2023, doi:10.3390/foods12040761_

Round 1

Reviewer 1 Report

The manuscript describes the use of liposomal encapsulation to improve the bitterness score of salmon frame protein hydrolysate and salmon frame protein plastein. While the authors have reported various measurements, most of the data is not well explained, supported by proper references and at times contradictory. Thus, this should be extensively revised. The followings are some specific examples. The authors should go over the entire manuscript and make substantial improvement.

1.      Are the SFPH and SFPT indeed encapsulated within the liposomes or absorbed on the exterior surface of the liposome, as the authors point out that the lower hydrophobicity of plastein allows it to interact with the phosphate (line 185)? This contradicts with the statement “positive charge of the hydrolysates which tended to neutralize the negative charge of the liposome, especially on the surface.”

2.      It is not clear how does the protein hydrolysates increase cohesion packing of the apolar domains in membrane vesicles. To be noted, the similar finding cited [25] is on essential oils, which have a different set of physicochemical properties. Can the authors provide some references and reconcile with the findings?

3.      The result section on ZP contains many sweeping statements without references or further explanation. E.g. “L-PT1 showed the highest negative charge. This was in agreement with the smallest size of the sample [line 233]”, “The smallest size was related with the larger surface area, allowing phosphate groups to be exposed to aqueous phase [line 234]”, “Usually, liposomes with more negative ZP had higher stability with the lower leakage… [250]”, and “Plastein products were found to impact the arrangement of liposome bilayer..”. Also, multilamellarity, as discussed is not supported by other measurements. In fact, all sections should be checked thoroughly to remove unsubstantiated statements.

4.      The TEM section is cryptic and difficult to understand. Perhaps, the authors can include a schematic to depict the arrangement of protein and lipids. Furthermore, the quality of the images need to be improved. The existing resolution does not support the discussion of bilayer, double layered, etc so this should be revised extensively.

5.     Table 1 and 2: (i) It is not clear what do the superscripts refer to in Table 1? More explanation is needed. (ii)The decimal places should be reduced. (iii) For Table 2, the surface hydrophobicity (SoANS) data seem to be missing.

6.      Line 38, “[3] found that the use of …” should be changed to “Sharma et al. found that… [3]”. This applies for all the other similar instances in the manuscript.

Author Response

Responses to reviewer

Comments to the author

The manuscript describes the use of liposomal encapsulation to improve the bitterness score of salmon frame protein hydrolysate and salmon frame protein plastein. While the authors have reported various measurements, most of the data is not well explained, supported by proper references and at times contradictory. Thus, this should be extensively revised. The followings are some specific examples. The authors should go over the entire manuscript and make substantial improvement.

*****Thank you so much for your invaluable time spent on our manuscript. All suggestions are taken into consideration for improvement of quality and clarity of our manuscript. All the corrections have been made using track changes.

  1. Are the SFPH and SFPT indeed encapsulated within the liposomes or absorbed on the exterior surface of the liposome, as the authors point out that the lower hydrophobicity of plastein allows it to interact with the phosphate (line 185)? This contradicts with the statement “positive charge of the hydrolysates which tended to neutralize the negative charge of the liposome, especially on the surface.”

*****Firstly, the majority of SPH and SFPT were encapsulated within the liposomes, whereas some portion might get adhered to the surface, which could be considered as free or unencapsulated hydrolysates or plastein. Hydrophilic portion of phospholipid was negatively charged caused by phosphate groups. Moreover, liposomes have polar heads on outer surface and towards the inner hydrophilic core after arrangement and the aforementioned heads had phosphate groups. Thus, plastein with lower hydrophobicity plausibly interacted with phosphate groups, localized inside the liposome. Nevertheless, some amount could also stick to the outer surface. In general, plastein obtained from salmon frame possessed lower hydrophobicity in comparison to hydrolysate. Thus, this characteristics might be associated with higher encapsulation efficiency of plastein.

Secondly, for the statement “positive charge of the hydrolysates which tended to neutralize the negative charge of the liposome, especially on the surface” refers to the unencapsulated free hydrolysate that tended to neutralize the charge on the surface. This could occur as observed by the encapsulation efficiency (EE) less than 100%. Also, the lesser EE was found for hydrolysates in comparison to that for plastein, Therefore, some peptides found in free hydrolysates might adhere or interact with the surface. As the consequence, the charge of the liposome could be modified.

To make the statement clearer and more understandable, some sentences have been rewritten (line 254-267, 260-262).

  1. It is not clear how does the protein hydrolysates increase cohesion packing of the apolar domains in membrane vesicles. To be noted, the similar finding cited [25] is on essential oils, which have a different set of physicochemical properties. Can the authors provide some references and reconcile with the findings?

*****Thank you for your insightful comment. We do agree that essential oil had different characteristics from protein hydrolysate or plastein. Thus, the mentioned phrase concerning the essential oil has been removed. However, the discussion based on reference no. 25 was still made, since it was related with lipophilic peptides in hydrolysates and plastein. Those peptides probably resulted in higher cohesion and packing among apolar chains in vesicular membrane, associated with the larger size of liposome (See line 209-216, 221-224, Figure 4).

  1. The result section on ZP contains many sweeping statements without references or further explanation. E.g. “L-PT1 showed the highest negative charge. This was in agreement with the smallest size of the sample [line 233]”, “The smallest size was related with the larger surface area, allowing phosphate groups to be exposed to aqueous phase [line 234]”, “Usually, liposomes with more negative ZP had higher stability with the lower leakage… [250]”, and “Plastein products were found to impact the arrangement of liposome bilayer.”. Also, multilamellarity, as discussed is not supported by other measurements. In fact, all sections should be checked thoroughly to remove unsubstantiated statements.

*****Thank you for your invaluable suggestions. The discussion has been checked and rewritten. The references have been provided to substantiate the statements (Please see lines 251-267, 271-275).

  1. The TEM section is cryptic and difficult to understand. Perhaps, the authors can include a schematic to depict the arrangement of protein and lipids. Furthermore, the quality of the images need to be improved. The existing resolution does not support the discussion of bilayer, double layered, etc so this should be revised extensively.

*****Thanks for your suggestion.  Authors have included a schematic to depict the arrangement of protein and lipids (Figure 4). In addition, the image for bilayer liposome has been replaced by the one with better resolution. Moreover, authors have rewritten some sentences for the better clarity (section 3.2.6, lines 476-493)

  1. Table 1 and 2: (i) It is not clear what do the superscripts refer to in Table 1? More explanation is needed. (ii)The decimal places should be reduced. (iii) For Table 2, the surface hydrophobicity (SoANS) data seem to be missing.

*****Thanks for the suggestion.

(i) Superscripts have been defined.

(ii) Decimal places have been reduced to 1 digit, except for PDI, in which 2 digits have been used.

(iii) We apologize for the typing error and mistake. The surface hydrophobicity was not determined for this experiment.

  1. Line 38, “[3] found that the use of …” should be changed to “Sharma et al. found that… [3]”. This applies for all the other similar instances in the manuscript.

*****Authors have taken your suggestion into consideration and have made the amendments accordingly throughout the manuscript. Thank you so much.

Reviewer 2 Report

The manuscript describes the preparation and characterization of liposomal formulations for protein derivatives encapsulation. The text is mostly clear and concise and may be of interest to the audience reached by the journal. Hence, I support minor changes, as stated below.   - Liposome vesicles, as written on line 44 of page 1, is redundant. Please, standardize the nomenclature. Also, multilamellar liposomes would be better named as multilayered instead of many layers. - Information on the TEM equipment should be provided. - Please, specify if any dilution was necessary for DLS measurements, assuming that the samples may be turbid. - Do the authors have any information on the long-term stability of the formulations? From zeta potential results, I suppose aggregation might be delayed, but this could be interesting to be explored. - It is not clear for me how the transition from multi to unilamellar liposomes, through sonication, causes changes in zeta potential values. Any reference to support it? - Minor english editing is recommended.

Author Response

Response to reviewer

Comments to the Author

The manuscript describes the preparation and characterization of liposomal formulations for protein derivatives encapsulation. The text is mostly clear and concise and may be of interest to the audience reached by the journal. Hence, I support minor changes, as stated below.

*****Thank you so much for your invaluable time spent on our manuscript. All suggestions are taken into consideration for improvement of quality and clarity of our manuscript. All the corrections have been made using track changes.

  1. Liposome vesicles, as written on line 44 of page 1, is redundant. Please, standardize the nomenclature. Also, multilamellar liposomes would be better named as multilayered instead of many layers. It needs to be improved in two points.

*****Thanks for your comment. Authors have made the changes as suggested (line 44 and 342).

  1. Information on the TEM equipment should be provided. - Please, specify if any dilution was necessary for DLS measurements, assuming that the samples may be turbid.

***** Thank you for the question. The information regarding TEM equipment has been added in section 2.5.2.4. (Line 163-167). Although the sample was turbid, no dilution was required for the measurement.

  1. Do the authors have any information on the long-term stability of the formulations? From zeta potential results, I suppose aggregation might be delayed, but this could be interesting to be explored (line 163).

*****Authors would like to thank the reviewer for the insightful suggestion.  Authors did not check the long term stability of the current formulations. Since our major focus was on debittering of hydrolysates using liposome. The stability study has been taken into consideration and the study will be carried out for the liposome loaded with hydrolysate as raised by the reviewer.  

  1. It is not clear for me how the transition from multi to unilamellar liposomes, through sonication, causes changes in zeta potential values. Any reference to support it?

*****Authors would like to inform that in the present study there was not much significant difference observed for zeta potential after sonication. The little difference was found in case of liposome loaded with protein hydrolysate, which might be due to the free peptides or hydrolysates (unencapsulated ones, since the EE is not 100%) that might be removed from the surface of the liposomes, induced by sonication. As a result, it caused the minor change of zeta potential. However, the EE was higher in encapsulated plastein. In that case, the sonication had the negligible effect on zeta potential. 

  1. Minor English editing is recommended.

*****Authors use the ‘Grammarly’ software for editing English in the manuscript.

Round 2

Reviewer 1 Report

It is regrettable to say that many of the revisions are unsatisfactory and the initial concerns are not resolved. Below is point-by-point response.

1. This is still confusing. Can the acetone method distinguish surface-attached, freely suspending (unattached) and entrapped portions? If not, then the EE calculation is misleading. This needs to be rectified.

2. As the authors have recognized the difference in properties between essential oil and the peptides, it comes at a surprise that the same reference is still used to support the cohesion and packing argument. This does not make sense to me.

3. The changes made are not sufficient to address the concerns. For instance, the statement, “This exposure of phosphate group towards the aqueous phase was also aided by the stabilizing agent, glycerol, as it has high content of hydroxyl groups”, is not supported by ref [2]. In ref [2], the effect of glycerol on phosphate group exposure has not been studied explicitly. Some other sentences have not been revised and substantiated.

4. The revised Fig. 3(c) and the text remain cryptic and unconvincing. The schematic diagram is helpful, but I am not sure if the results supports the depiction. The results are not explained in detail. I suggest to the authors to add, specifically, (i) statistics to the observation, (ii) explains what are the debris-like material surrounding the liposome in Fig. 5(a), (iii) explain why do the boundary of the “liposomes” in Fig. 5(b) not contiguous, (iv) if the darker interior reflect encapsulation in Fig. 5(c), why do the plastein-loaded ones in Fig. 5(b) appear empty, and (v) images of better quality are needed if encapsulation and bilayers are to be concluded and discussed. Moreover, it is not necessary to explain how a phospholipid self-assemble into a liposome. If needed, a more accurate explanation is needed. There is a much better reference that ref [29] for the associated statement. For instance, those by Bangham et al., Lipowsky et al., and Safinya et al., to name a few.

5. I am sorry to say this representation is confusing even with the note.

Additional comment:

Line 453: “Almost peptides (11pprox.. 95 %) were..”. The authors should proof read the manuscript carefully.

Author Response

Response to reviewer

It is regrettable to say that many of the revisions are unsatisfactory and the initial concerns are not resolved. Below is point-by-point response.

****Thank you for your time spent on the revision and providing the insightful comment for the clarity of manuscript. Sorry for the unclear responses. All the queries have been responded and the corrections have been made as appeared in the track change.

1.This is still confusing. Can the acetone method distinguish surface-attached, freely suspending (unattached) and entrapped portions? If not, then the EE calculation is misleading. This needs to be rectified.

***Acetone is considered as a polar aprotic solvent. However, ethanol has more polarity than acetone. Thus, acetone could extract the peptides in hydrolysate or plastein, while the phospholipid was still present in the liposome formed. Therefore, acetone was selected to extract the free or untrapped peptides.

For surface-attached peptides, some of them might be extracted, depending on how strong the interaction between peptides and the surface or wall of liposome was. Sorry. Authors did not check the peptide at the interface as raised by the reviewer. However, it will be taken into consideration for our future work since it is the invaluable comment.

  1. As the authors have recognized the difference in properties between essential oil and the peptides, it comes at a surprise that the same reference is still used to support the cohesion and packing argument. This does not make sense to me.

****Authors totally agree with the reviewer for the differences in nature between essential oil and peptides served as the core in liposome. However, some sentences regarding the general concept was still kept in the text in the former version. To avoid the confusion and contradiction, all the sentence from the aforementioned reference has been removed from the manuscript. Sorry for those non-scientifically sound statements.

  1. The changes made are not sufficient to address the concerns. For instance, the statement, “This exposure of phosphate group towards the aqueous phase was also aided by the stabilizing agent, glycerol, as it has high content of hydroxyl groups”, is not supported by ref [2]. In ref [2], the effect of glycerol on phosphate group exposure has not been studied explicitly. Some other sentences have not been revised and substantiated.

****Authors agree with the reviewer that some sentences in reference number 2 was not substantiated. Therefore, we removed those statements from the text. For other sentences, we have cross checked and substantiated. Thankyou so much for this invaluable comment.

  1. The revised Fig. 3(c) and the text remain cryptic and unconvincing.

****The TEM image can prove that there is the wall of spherical liposome containing the core as shown in Fig. 3(c). Similar results were also reported by Tagrida et al. (2021), Chotphruethipong et al. (2020), the images of which has been shown below.

Fig: TEM image of loaded liposome (Ref:Tagrida et al., 2021)

Tagrida, M., Prodpran, T., Zhang, B., Aluko, R.E. and Benjakul, S., 2021. Liposomes loaded with betel leaf (Piper betle L.) ethanolic extract prepared by thin film hydration and ethanol injection methods: Characteristics and antioxidant activities. Journal of Food Biochemistry, 45(12), p.e14012.

Fig: TEM image showing bilayer vesicle (Ref: Chotphruethipong et al., 2020)

(Chotphruethipong, L., Battino, M. and Benjakul, S., 2020. Effect of stabilizing agents on characteristics, antioxidant activities and stability of liposome loaded with hydrolyzed collagen from defatted Asian sea bass skin. Food chemistry, 328, p.127127.)

The schematic diagram is helpful, but I am not sure if the results supports the depiction. The results are not explained in detail.

****We do agree with the reviewer and admit that the schematic diagram was drawn for the possible localization of peptides at surface, the core and between the bilayer. The advanced analytical instrument must be required to elucidate the proposed scheme. Unfortunately, we did not have very advanced instrument at our university. However, the schematic diagram can make the readers understand much more as suggested by the reviewer for the first revision.

I suggest to the authors to add, specifically, (i) statistics to the observation,

****We did not make any comparison in term of quantitative analysis. Thus, we do not know what kind of statistics must be used. Sorry for this.

 (ii) explains what are the debris-like material surrounding the liposome in Fig. 5(a),

****The debris surrounding the liposome was probably the free phospholipids, which was distributed in the free form, not in the form of bilayer. The discussion was included in the text. Please see line 482-484.

(iii) explain why do the boundary of the “liposomes” in Fig. 5(b) not contiguous,

**** This is because most liposomes were negatively charged. Therefore, the repulsion between different liposomes took place, thus keeping those liposomes separated from each other.  The explanation has also added in the manuscript. Please see line 485-488.

 (iv) if the darker interior reflect encapsulation in Fig. 5(c), why do the plastein-loaded ones in Fig. 5(b) appear empty,

****The different magnifications were used between those two figures. The contrast was adjusted to see the outer layer of liposome along with the higher magnification in Fig. 5c. Overall, the image brightness generally decreased as magnification increase. That’s the reason why the core was dark in Fig. 5c. Explanation has been given in line 488-491.

and (v) images of better quality are needed if encapsulation and bilayers are to be concluded and discussed. Moreover, it is not necessary to explain how a phospholipid self-assemble into a liposome.

****The images presented in the current study are the best we can obtain from the instrument we have. Sorry for this. However, it can represent the liposome structure, which have been well known.

As suggested, the explanation how the phospholipid arrange itself into liposome has been removed since it has been well elucidated for a long time.

If needed, a more accurate explanation is needed. There is a much better reference that ref [29] for the associated statement. For instance, those by Bangham et al., Lipowsky et al., and Safinya et al., to name a few.

****Following the reviewer’s suggestion, the explanation has been removed. So, authors decided not to extend the explanation on self-assembly of phospholipids to liposome.

  1. I am sorry to say this representation is confusing even with the note.

****Sorry once again. With the help from reviewers to raise point to point on the unclear parts, the manuscript has been improved with no confusion. If ‘this representation is confusing’ means ‘unclear schematic diagram’, authors used the general concept of liposome structure to represent the aforementioned diagram and the location of the peptide was most likely based on their polarities as shown in the diagram. Authors are grateful for the invaluable and insightful comments given by the reviewer.

Additional comment:

Line 453: “Almost peptides (11pprox.. 95 %) were..”. The authors should proof read the manuscript carefully.

****Sorry for the mistake. Authors have corrected and checked throughout the manuscript.
